# Explorative Supercooling Technology for Prevention of Freeze Damages in Vaccines

**Shawn Jun [1], Youngbok Ko [2],\* and Seung Hyun Lee [3],\***

[1]  Jun Innovations Inc., 2800 Woodlawn Dr. Suite 298, Honolulu, HI 96822, USA; sjun@juninnovationsinc.com
[2]  Department of Obtetrics and Gynecology, College of Medicine, Chungnam National Uninversity, Daejeon 35015, Korea
[3]  Department of Biosystems Machinery Engineering, College of Agriculture and Life Science, Chungnam National University, Daejeon 34134, Korea
\*  Correspondence: koyoung27@cnuh.co.kr (Y.K.); seunglee2@cnu.ac.kr (S.H.L.); Tel.: +82-42-580-8107 (Y.K.); +82-42-821-6718 (S.H.L.); Fax: +82-42-823-6246 (S.H.L.)

**Abstract:** Most freeze-sensitive vaccines are stored between 2 °C and 8 °C upon manufacturing and until they are eventually administered in intermediate vaccine stores and health facilities. This so-called "cold chain" of vaccine distribution is strictly regulated at these specific temperatures to avoid freeze damage. Liquid formulations of particular vaccines (e.g., aluminum-adsorbed tetanus toxoid (TT)) will irreversibly lose their immunogenicity once frozen. Using an oscillating magnetic field (OMF), supercooling can inhibit ice crystal nucleation effectively; water is susceptible to influence by a strong magnetic field, allowing normal water dynamics even in subzero freezing conditions. This recently developed technology—composed of a custom-designed electromagnet unit producing an optimal field strength (50 mT) at a specific frequency (1 Hz)—was successfully used to inhibit the formation of ice crystals in aluminum adjuvant TT vaccines, therefore preventing any visible damage in the vaccines' microscopic structure. Despite being subject to temperatures far below their freezing point (up to −14 °C) for up to seven days, the TT vaccines showed no freeze damage on physical appearances. Results were further validated using shake tests and light microscopy. As storage and freeze-protection become more critical during times of increased vaccination efforts—particularly against COVID-19—this supercooling technology can be a promising solution to distribution problems by removing concern for temperature abuse or shock-induced freezing.

**Keywords:** freeze-sensitive vaccine; temperature abuse; magnetic field; subzero temperature; cold chain



## 1. Introduction

Vaccines are medical tools that enable people to acquire partial or complete immunity against harmful and virulent pathogens. Historically, macroscale examples from our modern society exemplified the great speeds at which widespread vaccination had hindered and offset the spread of viruses and bacteriophages. With vaccines, more than three million lives are saved annually from death caused by pathogens [1]. Vaccines have transformed modern medicine, becoming incredible and effective measures against outbreaks of disease. The Center for Disease Control and Prevention (CDC) recommends an eighteen-year immunization schedule for newborn infants that provides comprehensive immunity against a myriad of disease-causing viruses and bacteria [2].

Despite the importance of vaccines, several problems have emerged and embattled efforts to promote vaccination. Some issues, such as vaccination hesitancy and misinformation, are directly influenced by human actions, while other problems—storage complications in particular—arise due to the inherent nature of vaccines. A patent drawback on most vaccines is their rather characteristic requirements in storage temperature. This is

because temperature damage can irreversibly lessen vaccines' effectiveness and immunogenicity, which often necessitates the destruction of damaged stocks. The World Health Organization (WHO) suggests that all vaccines except the oral polio vaccine be stored in the temperature range from 2–8 °C [3]. To that end, vaccine storage is generally monitored under strict standards, and vaccines are kept at recommended temperatures for as long as possible.

Vaccines are often heat-protected but may suffer greater damage from freezing. Indeed, certain vaccines—but not their diluents—can be stable in subzero temperatures as per manufacturers' and the WHO's guidelines, but freezing is seldom essential and rarely recommended in any case. Vaccines containing aluminum adjuvants are irreversibly damaged when frozen due to the clumping of adjuvants, eliminating the immunological properties of the vaccine [4]. This can be a significant bottleneck in our cold chain, considering aluminum adjuvants are the most common adjuvant type used in human vaccines: diphtheria, tetanus, pertussis, liquid *Haemophilus influenzae* type b (Hib), hepatitis B, and inactivated poliovirus vaccines are all examples of the aluminum adjuvant type [5]. Today, vaccines are stored in refrigerators or well-insulated cold boxes with deep-frozen ice packs [6]. Deep-frozen ice packs can reach temperatures as low as −20 °C, which poses a high freezing risk to the vaccines stored in close proximity to these packs.

Other common causes of vaccine freezing include inappropriate usage and placement of thermostat containers, misusage of refrigerators, misplacement of vaccines within, and inadequate temperature monitoring. While the WHO provides guidelines to mitigate vaccine freezing—such as conditioning ice packs and monitoring temperatures twice every 24 h—they are often prone to human error or ignored. It was found that during transport, freezing temperatures occurred 35.3% of the time in developing countries and 21.9% of the time during storage [5]. This problem has been well documented; it has been reported that in Mongolia, administration of the hepatitis B vaccine during the winter months is associated with poor vaccination effectiveness [7]. In the United States, regions with a higher percentage of refrigerators with frozen temperatures recorded a higher incidence of pertussis [8].

Current methods of preventing vaccine freezing include, but are not limited to, constant temperature monitoring, high precision refrigerators, the development of freeze-resistant vaccines, and the use of phase change material (PCM) packs. The WHO has also published an official guide on how to prevent vaccine freezing. Guidelines include correct placement of vaccines within storage units, using water packs instead of ice packs to pack vaccines, and recording temperatures twice every 24 h [6]. However, it has been found that the WHO's recommendations have often been ignored [5]. The costs of the said methods can be prohibitively high, and there is an apparent lack of a cost-effective solution. PCM packs, freeze-resistant vaccines, and more accurate refrigerators are all costly and admittedly nonefficient.

While mRNA vaccines are not of similar composition to those of the aluminum adjuvant type, remarkably unprecedented research during the current pandemic has been conducted to enable their expedient development, manufacturing, and shipping. Such research in bioinformatics, particularly research on the topic of genetic sequencing, has provided avenues for the development of a new class of vaccines. mRNA vaccines, due to their direct targeting of a virus's genetic and molecular structures, may prove to be a boon for the future of modern medicine. However, perhaps due to the novelty of these vaccines, their storage requirements are not a well-discussed matter at all. Early batches for trials were kept at −70 °C by default. Various recommendations for storage temperature were eventually put forth for various prototypes and specific mRNA profiles; they were usually ultracold temperatures. For much of the development and initial production of these vaccines, ultracold storage was a dire yet perpetually unsatisfied need. More concerningly, long-term storage of mRNA vaccines was never intensely studied; many studies only reported short-term (six months) stability in vaccines kept in refrigerated conditions [9,10] While bioinformatics has already proved its advantages in medicine development, basic

needs must be met—such as proper storage—so that research can be done in an efficient yet effective manner.

Currently, fully-developed mRNA vaccines against SARS-CoV-2 are stored in particular and often fastidious ways per the guidelines of their respective manufacturers. Some currently available vaccines are stored at extremely low temperatures before mixing with their respective diluents. For example, the Pfizer-BioNTech vaccine is only stable for ten days in dry-ice-supported thermal shipping containers. After shipping, the vaccine can be stored at ultracold temperatures (−60 °C to −90 °C) but only for two weeks. After this period, the vaccine must be stored in refrigerator conditions (2 °C to 8 °C) for a maximum of 31 days [11]. The Moderna vaccine is, indeed, stable at (−15 °C to −50 °C) until its expiration date, but unused vaccines last for up to 30 days in a refrigerator [12]. In any case, ultracold freezing is never recommended as a long-term storage method. Instead, refrigeration offers a longer storage term with no risk of reduced potency. Though not entirely attributed to temperature abuse or damage, SARS-CoV-2 vaccines are often thrown out by the tens of thousands due to breakage, storage and transportation problems, and expiration [13]. A cold storage option covering all three conditional needs (thermal shipping, ultracold storage, and refrigeration upon thawing) can significantly benefit the vaccination effort against COVID-19 by boosting efficiency and distribution simplicity.

During the late 1980s, a reliable "shake test" method to detect vaccine freezing was developed. Empirical observations in the field found visible differences between frozen-then-thawed vaccines and never-frozen vaccines. Upon freezing and thawing of a vaccine sample, the lattice structure composed of the vaccine's adsorbent (i.e., aluminum) and antigen breaks. The adsorbent, which becomes significantly heavier as its particulates separate, gradually settles at the bottom of the vaccine sample minutes after shaking the sample. Additional repetitions of freezing and thawing the vaccine sample are more likely to increase the sizes of adsorbent granules. These granules form a visibly clear supernatant while also forming white sedimentation at the bottom of the sample container [14]. In contrast, undamaged aluminum-adsorbed vaccines remain as white, homogeneous solutions even after shaking.

Kartoglu et al. (2010) confirmed the value and validity of the shake test by proving freeze damage in aluminum-based vaccines. They designed a double-blind crossover model to compare the performance of the shake test conducted by trained workers with visual outcomes of phase-contrast microscopy. A total of 475 vials of eight different WHO prequalified freeze-sensitive vaccines from ten distinct manufacturers were used. Reporting 100% sensitivity, specificity, and positive predictive value in the shake test, the study provided confidence that the test could promote proper handling by vaccine dispensers [10]. Click or tap here to enter text. However, the test is not a preventive measure against vaccine temperature abuse; it may only detect previously frozen vaccines so that they are not administered to patients.

Supercooling is a currently explored method whereby subzero storage of temperature-sensitive and water-based products—including vaccines—is possible without the temperature damage caused by freezing $H_2O$. By definition, supercooling is a process of cooling biomaterials below their freezing points without ice crystal nucleation. Several innovative and practical approaches have been implemented to achieve stable and acceptable reproducibility of this supercooling phenomena in biological samples; such examples include surface sealing by oil and alcohol phases [15], high pressure [16,17], ultrasound [18], and electrostatic fields [19]. However, these applications were limited to either fundamental exploration or food applications.

Deep supercooling associated with biomedical applications studied by Usta et al. [20] removed the storage medium–air interface using an immersible phase. The team sealed the surface of a small water sample (1 mL) with a hydrocarbon-based oil, (e.g., mineral oil, olive oil, and paraffin oil) that blocked ice formation at subzero temperatures for up to a week. By experimenting with more complex oils and with pure, simple hydrocarbons, such as alcohols and alkanes, they were able to keep a small volume of water and cell suspension

(1 mL) supercooled at −20 °C for 100 days and a larger volume (100 mL samples) for a week.

A new supercooling technique for human organs was aimed to extend the period from the moment of extraction to 1.5 days rather than 5–10 h, providing a valuable window of time during which patients can obtain viable organs from donors. The invention perfused a liver with antifreeze chemicals to lower the freezing point below 0 °C and then accomplished uniform cooling throughout the organ, allowing the organ sample to hibernate or enter suspended dynamism [21]. However, the current state of their technologies has not matured enough to be widely adopted in commercial applications.

The supercooling technology in this study utilizes a controlled, oscillating magnetic field to prevent freezing within biomaterials (Figure 1a). It is theorized that the magnetic field vibrates and disrupts the bonds between water molecules that form during ice crystal nucleation, thus preventing the nucleation of water. This electromagnetic technology—combined with pulsating electric fields—has already demonstrated its prevention of ice formation in meat and fruit products subject to subzero temperatures [22]. We believe that this technology can therefore be extended to preserve aluminum adjuvant vaccines and their biomaterials within. This study aimed to test the OMF-based supercooling technology in prevention of freeze damages of vaccines at subzero temperatures by comparing results with the aforementioned shake test and light microscopy.

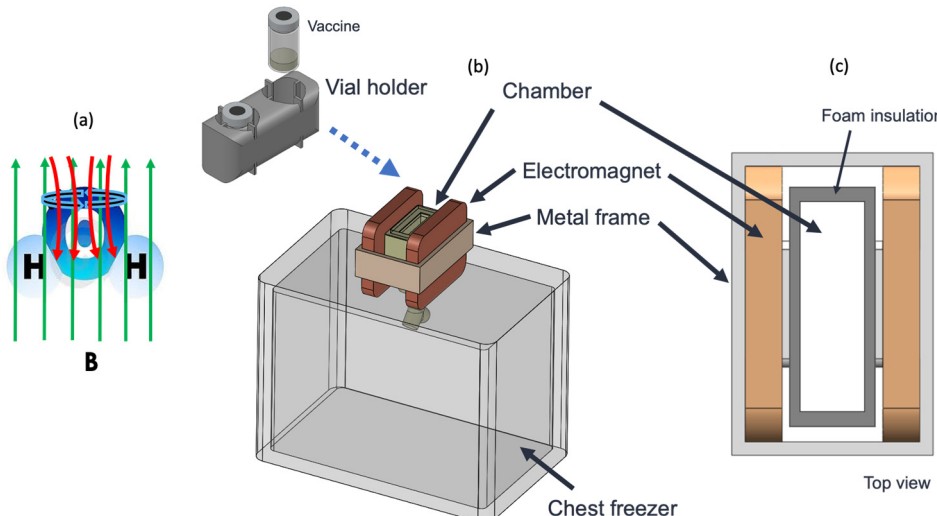

**Figure 1.** (**a**) Vibrating effect on water molecules (H$_2$O) due to the oscillating magnetic field *B*, (**b**) a schematic of the experimental setup including the external supercooling chamber, vaccine sample holder, and electromagnet, and (**c**) top view of the supercooling chamber with foam-insulated walls.

## 2. Materials and Methods

### 2.1. Design of the Supercooling Unit

The current supercooling chamber (150 mm (W) × 36.7 mm (D) × 122 mm (H)) was designed and fabricated as Figure 1b. The exterior cooling chamber was designed to circulate cold air from the freezer into the chamber via piping components. An OMF was produced using a custom-built electromagnet with variable magnetic field intensities up to 100 mT based on a 1-L-volume treatment chamber. The electromagnet was mounted on the top of a 5.3 ft$^2$ chest freezer (ZXS-150, Alamo Refrigeration, San Antonio, TX, USA) for thermal isolation from the controlled temperature of the freezer. The electromagnet was operated with a custom-designed power supply. An alternating current (AC) with a square waveform at the desired frequency was provided using a function generator (33220A, Agilent Technologies, Santa Clara, CA, USA). The intensity of the applied OMF was measured using a Teslameter (F71, Lake Shore Cryotronics Inc., Westerville, OH, USA). A uniform magnetic field across the entire chamber are critical to ensure supercooling stability. All chamber components were fabricated to have an inner void space, and the

remaining gap was filled with spray foam insulation to minimize heat gain and loss from the ambient temperatures (Figure 1c). The ambient temperature of the freezer was regulated by a PID controller (D1S-2R-220, SESTOS Electronics H.K.) to achieve a desired temperature for the chamber. The temperatures of vaccine samples were monitored every 10 s using a data acquisition unit (Agilent 39704A, Agilent Technologies, Inc., Santa Clara, CA, USA).

### 2.2. Testing Vaccines in the Unit for Freeze Prevention

Aluminum-based freeze-sensitive vaccines, Tetanus Toxoid (TT) Livestock Vaccine (1 mL/vial, #40276), were acquired from online vendor Valleyvet.com (www.valleyvet.com, accessed on 16 April 2021). For a total of 8 vials per trial, a pair of vials were stored in each of the following conditions: (1) supercooling unit at −14 °C; (2) standard freezer at −20 °C; (3) refrigerator at 7 °C; and (4) supercooling unit at −14 °C without an applied OMF to serve as a negative control. Vaccines were stored for 24 h, removed, and allowed to thaw if necessary. One of the two supercooled vials (and one control vial) was left untouched for 7 days to observe long-term storage. The vials were labeled accordingly and compared using the shake test to determine whether ice formation had occurred during the trials. Observations were made 15 min after the shaking and were then photographed. These trials were repeated until we were certain beyond reasonable doubt (>95% confidence interval) that the supercooling chamber could prevent ice crystal formation in vaccines at −14 °C (Figure 2).

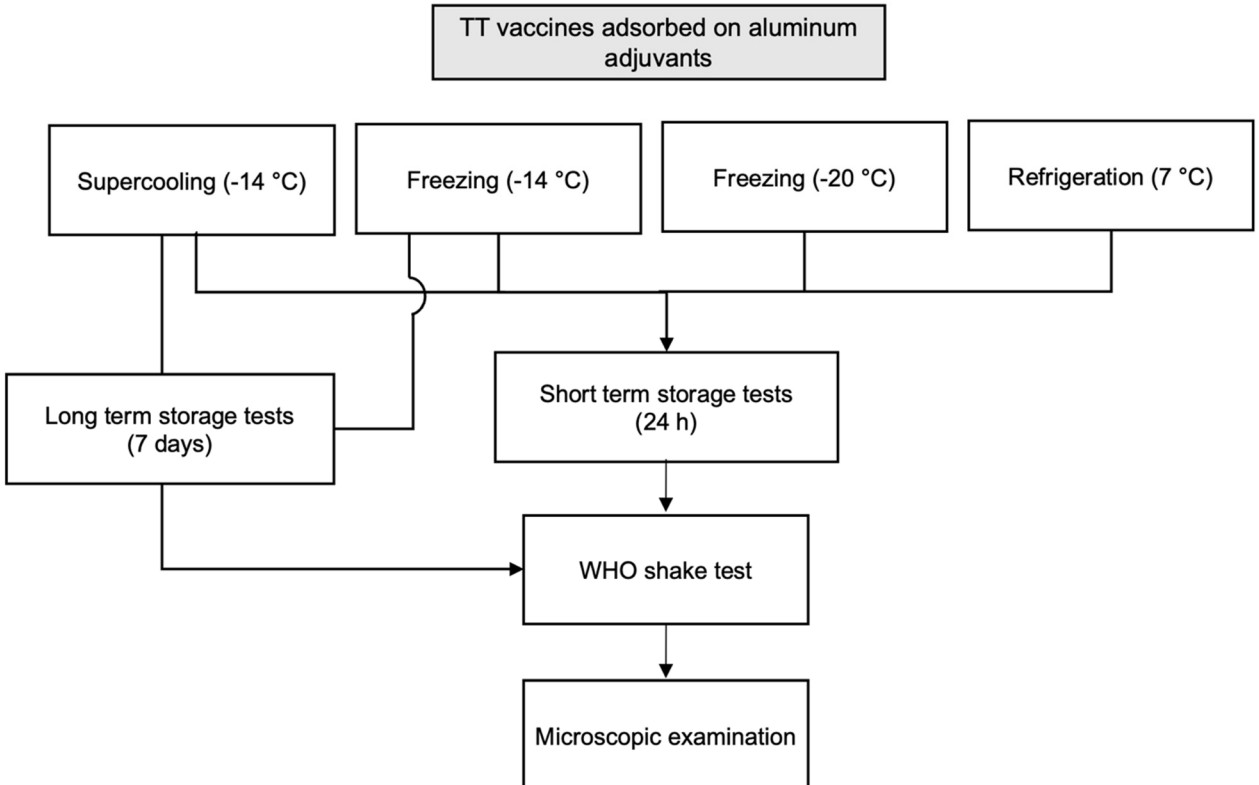

**Figure 2.** A flow chart of the experimental design to validate the supercooling preservation of vaccines.

### 2.3. Light Microscopy

All study samples were examined by DIC (differential interference contrast) with an Olympus BX-51 light microscope (Leica Microsystems Inc., Buffalo Grove, IL, USA). Images were taken with a Leica DFC7000T digital camera. After each vial was vigorously shaken, 10 μL of the vaccine in each vial were dropped onto a slide using a syringe, and a coverslip was placed over the sample. All samples were examined for structural formations under

$100\times$ magnification and were photographed under $10\times$ and $40\times$ magnifications. All snapshots were digitized, and particle size was measured. Results were coded numerically for "supercooled", "frozen", and "nonfrozen" vaccines (Figure 2).

## 3. Results and Discussion

Supercooling operating parameters such as field strength, frequency, duty cycles, and sample volumes (10 mL as a default vial volume) were tested and optimized. It was previously found that the probability of supercooling over long periods was significantly increased with OMF intensities of 50 mT. Therefore, OMF strength was optimized at 50 mT, 60 V, and 1 Hz with a 1 s on/off cycle. It should be noted the Food and Drug Administration (FDA) in 1987 limited magnetic field magnitudes on biomaterials at a maximum of 2000 mT, while our projected OMF strength was exercised at 50 mT [23]. Figure 3 shows that the supercooled vaccine was successfully stored at $-14$ °C, and the negative control without OMF treatment experienced the phase transition at 2.5 h into the trial and froze. Temperature profiles for other control vaccine samples (freezer at $-20$ °C and refrigerator at 7 °C) showed unremarkably expected results, with samples freezing in the freezer and remaining unfrozen in the refrigerator.

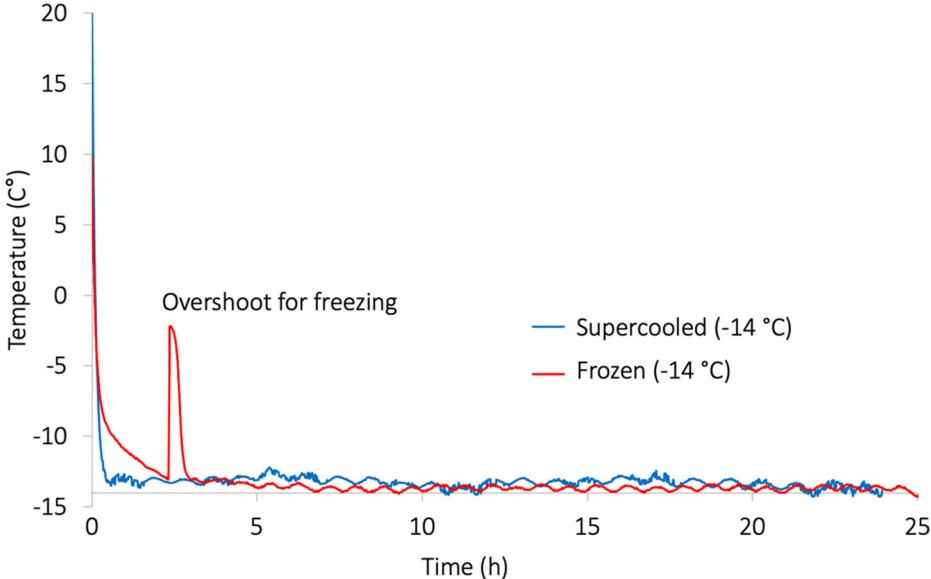

**Figure 3.** Temperature profiles of supercooled and control vaccines. The control vaccine sample (without OMF treatment) was frozen after 2.5 h of storage as indicated by the temperature overshoot. In general, as the temperature of the sample is lowered below its freezing point, ice nucleation occurs. This phase change causes the release of latent heat with the freezing point being revealed. Once all latent heat has been released, the temperature of the sample continues to decline until it matches the temperature of its ambient surroundings.

The shake test was designed based on the apparent differences in sedimentation rates of previously frozen and unfrozen vaccines to understand whether freeze-sensitive vaccines are damaged by freezing. Figure 4 shows the appearance of sample vials 3 min after they were vigorously shaken; these vials were previously stored at four different conditions for 24 h. Test vials (a) and (b) were stored at $-20$ °C and $-14$ °C, freezing in their respective storage units and then thawed before commencing the shake tests. Vials (c) and (d) were left unfrozen in their respective conditions, with vial (d) being subject to supercooling treatment during the 24 h. Sedimentation in frozen-then-thawed vaccines was quick; a definite, clear supernatant formed gradually above a white disk of coagulates at the bottom. However, much similar to the refrigerated test vaccine, the supercooled vials showed no sedimentation and remained a cloudy, homogeneous mixture after the shaking. It is known that ice crystals formed during freezing force aluminum particles

to overcome repulsion, thereby producing strong interparticle attraction. This ultimately results in aluminum particle coagulation and agglomeration [10]. Consequently, these particles become heavier and sediment faster in their respective suspensions.

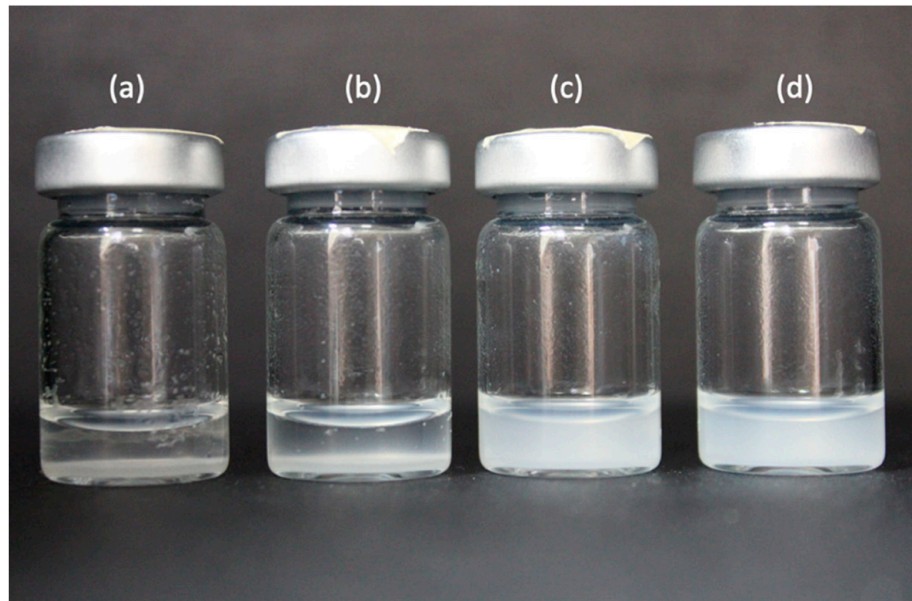

**Figure 4.** Snapshots of supercooled and control vaccines after shake tests. Test vials (**a**,**b**) were stored frozen at −20 °C and −14 °C (without OMF treatment) for 24 h, and then thawed before commencing the shake tests. Vial (**c**) was refrigerated at 7 °C and vial (**d**) was subject to supercooling treatment for 24 h.

Figure 5 shows microscopic snapshots of supercooled vaccines vs. other controls under light microscopy. Freeze-damaged vaccine samples (i.e., frozen (−20 °C) and frozen (−14 °C)) appeared to consist of large conglomerates of massed precipitates with amorphous, crystalline, solid, and needlelike structures; however, the supercooled vaccine showed fine grain structures (in both 10× and 40× magnifications), which is identical to vaccine stored under the refrigeration (7 °C). Aggregates in frozen vaccines measured up to 500 μm and 350 μm on average. Particles in undamaged TT vaccine samples were measured at 2 and 20 μm. The concordance in establishing the status of a TT vaccine as frozen or non-frozen was 100% between the light microscopy and the shake test method. Therefore, the vaccines kept supercooled at −14 °C were not freeze-damaged and most likely retained their full potency and immunogenicity during storage.

A TT vaccine was supercooled for 7 days to test if the vaccine's physical characteristics could be maintained for an extended period. In Figure 6, the supercooled was successfully preserved by OMF treatment without any phase transition for 7 days. A shake test was conducted and compared the supercooled vaccine with the control vaccine (Frozen (−14 °C)). The test results indicated a significant difference in the sedimentation rates between the supercooled and frozen vaccines. In general, TT vaccines adsorbed on aluminum adjuvants are stable during the controlled cold supply chain. However, they may change their appearance and lose potency when frozen because the adjuvant gel structure is destroyed by freezing.

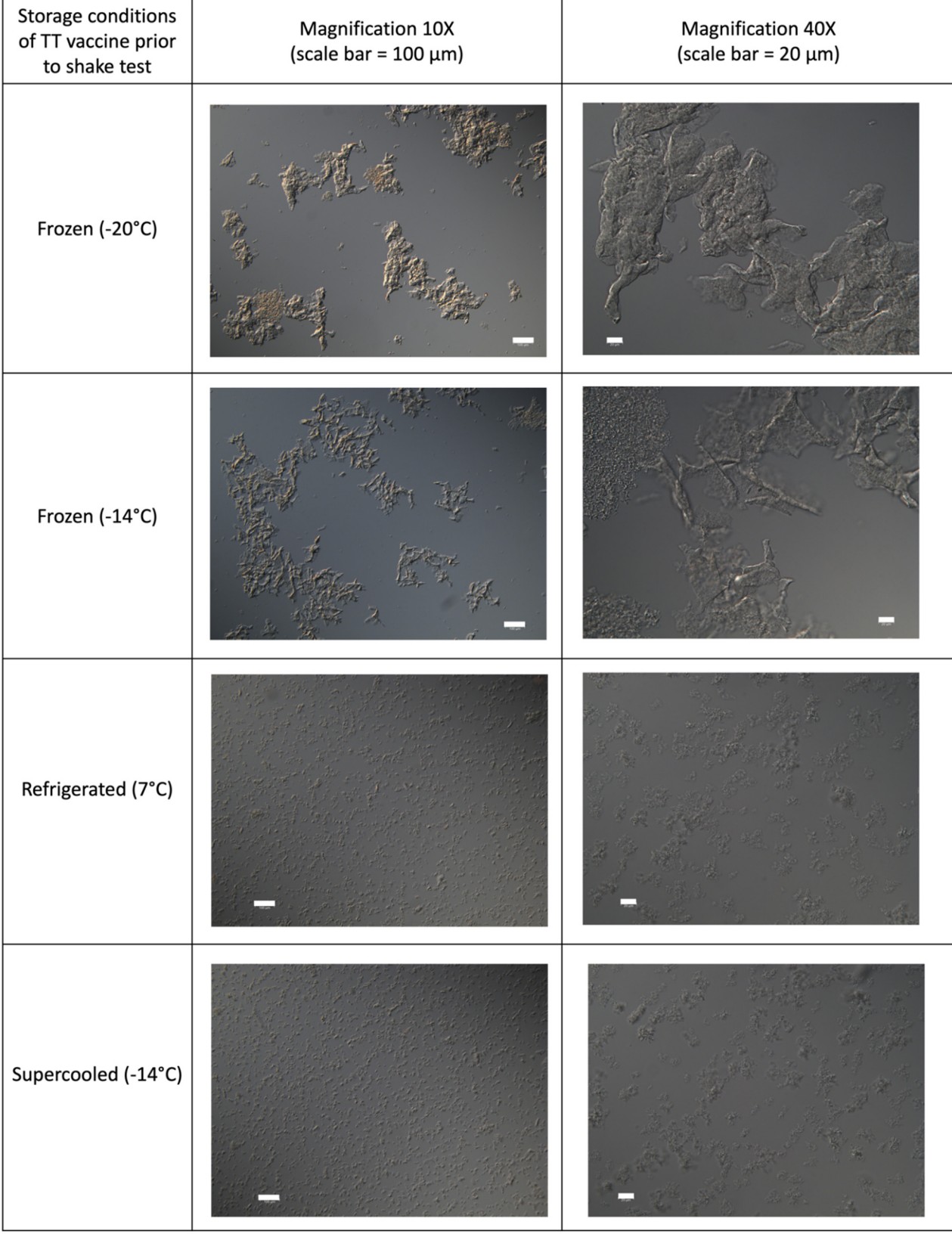

**Figure 5.** Microscopic views of supercooled (−14 °C) and other control vaccines (frozen (−20 °C), frozen (−14 °C), and refrigerated (−20 °C)) after shake tests. Scale bars indicate 100 μm and 20 μm for magnification 10× and 40×, respectively.

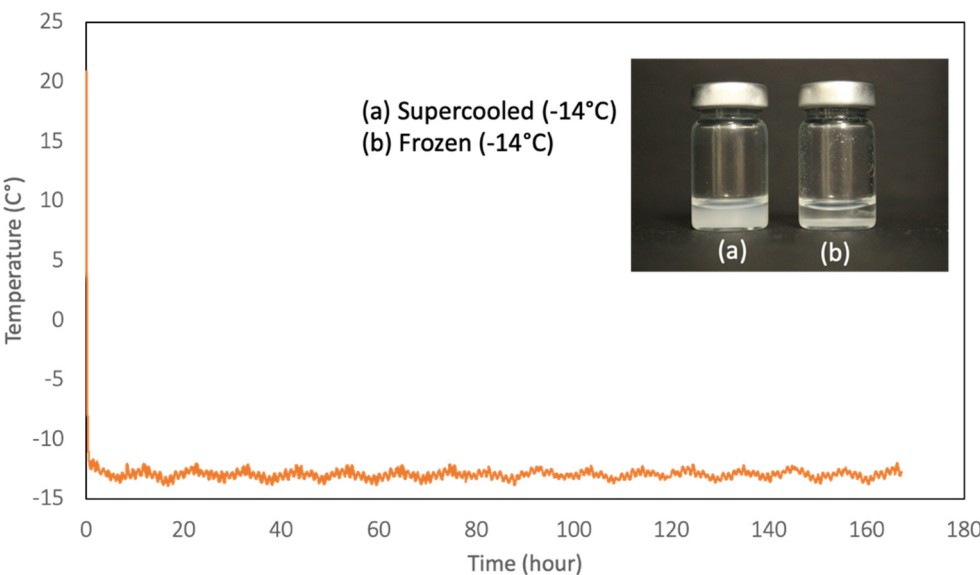

**Figure 6.** Temperature profiles of study vaccines after 7 days of storage: (**a**) supercooled ($-14\,°C$) and (**b**) frozen ($-14\,°C$). Inserts show snapshots of supercooled and control vaccines after shake tests.

Real efficacy data are often difficult to obtain, as each product has its particular threshold for freeze damage. As such, there may be a difference between exposure to freezing temperatures and freezing a vaccine sample sufficiently to destroy its potency. Furthermore, the shake test's inability to distinguish between exposure to freezing temperatures and loss of potency due to freezing may create a lack of trust in its veracity. However, as seen with all the control samples in this study's different trials, both short- and long-term exposure to subzero temperatures resulted in freezing, proving these temperatures were well below the freezing point. Additionally, the WHO assures that freezing at these temperatures would have, without doubt, resulted in a loss of potency for aluminum adjuvant TT vaccines [3]. On the other hand, the supercooled vaccines treated by OMF for 7 days passed the shake test and did not experience any ice crystal nucleation. Therefore, it is reasonable to conclude that the developed supercooling technology can ensure the physical stability of freeze-sensitive vaccines during storage and transportation for prolonged periods of time, perhaps paving the way to improve the potency.

## 4. Conclusions

The OMF-based supercooling innovation has no detrimental impact on physical characteristics of freeze-sensitive vaccines, even when they are stored at temperatures below freezing point. Therefore, this technology can also eliminate the likely human error associated with varying instances of freeze damage in vaccines. Indeed, guidelines set forth by WHO are unfollowed and have shown to be ineffective in preventing temperature abuse among vaccine distributors and dispensers [3]. It can be envisioned that the supercooling technology becomes an integral feature in the "cold chain", regardless of storage specifications and temperature-related needs for different vaccines.

As long as a power supply is available at every part of vaccine distribution, workers would simply place the vaccines in a storage container with supercooling technology. Maintenance of supercooling units would be minimal, and careful storage of vaccine products with ice packs would not be needed. Furthermore, separating diluent and pellets would no longer be necessary for such aluminum adjuvant vaccines as the *Haemophilus influenzae* type b (Hib) vaccine, as reconstituted vaccines could be stored without much worry for accidentally freezing the diluent liquid or damaging the vaccine itself [24].

Perhaps the developed supercooling technology, further assisted by bioinformatics, could provide an umbrella option where no concern for the various types of storage and temperature specifics (thermal shipping, ultracold storage, and refrigeration upon thawing)

is necessary. In this sense, supercooling would act as a subzero storage situation for these vaccines without the drawbacks of freezing the vaccines and eventual thawing.

Future work will definitely include the enzyme-linked immunosorbent assay (ELISA) and immunogenicity tests based on animal models to validate the supercooling preservation. Thereafter, commercially viable units can be designed in consideration of the most optimal magnetic field strength, uniformity, and application frequency specifications for various vaccines—not only aluminum adjuvant types—and then designing for macroscale, larger-capacity supercooling chambers. Electromagnet design types (ferric-core-material-based, Helm-Holtz-based, etc.) and their limitations will heavily depend on the chamber's established specifications, with practical limitations set upon the weight, design, shape, and volume of said chamber. Despite its novelty and need for in vivo validation of immunogenicity, the supercooled vaccine storage can potentially be a promoter of increased vaccine distribution and usage in this field.

In addition, cryopreservation has been successful in numerous cell types and some simple tissues. However, conventional approaches to cryopreservation cannot be applied to more complex natural or engineered multicellular tissues due to the destructive effect of extracellular ice formation. Restricting the size and extent of ice crystal formation during cryopreservation was marginally achieved using sufficiently high concentrations of cryoprotectants to promote amorphous solidification (vitrification) rather than crystallization. The supercooling technology is expected to be an alternative to cryopreservation technology for mammalian cells, organs, and tissues by lessening the toxicity issue of cryoprotectants and simplifying the procedure.

**Author Contributions:** Writing—original draft preparation, S.J.; investigation, S.J.; writing—review and editing, Y.K. and S.H.L.; project administration, S.H.L. All authors have read and agreed to the published version of the manuscript.

**Funding:** This research received no external funding.

**Institutional Review Board Statement:** Not applicable.

**Informed Consent Statement:** Not applicable.

**Data Availability Statement:** Data presented in this study are available in the article.

**Conflicts of Interest:** The authors declare no conflict of interest.

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
