# Peer review of "Explorative Supercooling Technology for Prevention of Freeze Damages in Vaccines"

_applsci, doi:10.3390/app12063173_

Round 1
Reviewer 1 Report
The manuscript “Explorative Supercooling Technology for Prevention of Freeze Damages in Vaccines”, by Jun et al, describe the application of an oscillating magnetic field to inhibit ice crystal nucleation in vaccine formulations. The authors argue that this method can be a very promising solution to distribution problems by removing concern for temperature abuse or shock-induced freezing. Eventhough, immunogenicity data is lacking, the study was well conducted and well presented. I have no major concerns about this manuscript.
Minor concerns/comments:
1) The authors propose this technology could potentially be applied to COVID-19 vaccines. Given that the mRNA COVID-19 vaccines are stored at -80º C, could the authors please comment on the possibility of applying this technology to freeze at these temperatures?
2) The authors tested the longevity of this method for 7 days only. Could the authors provide additional data about longer vaccine storage using this method?
3) Finally, could the authors comment on the potential application of this method to cryopreserve mammalian cells?
Author Response
The manuscript “Explorative Supercooling Technology for Prevention of Freeze Damages in Vaccines”, by Jun et al, describes the application of an oscillating magnetic field to inhibit ice crystal nucleation in vaccine formulations. The authors argue that this method can be a very promising solution to distribution problems by removing concern for temperature abuse or shock-induced freezing. Even though immunogenicity data is lacking, the study was well conducted and well presented. I have no major concerns about this manuscript.
Minor concerns:
- The authors propose this technology could potentially be applied to COVID-19 vaccines. Given that the mRNA COVID-19 vaccines are stored at -80º C, could the authors please comment on the possibility of applying this technology to freeze at these temperatures?
Agreed. The following paragraphs were added to Page 3.
“For example, the recently FDA-approved Pfizer-BioNTech vaccine is only stable for ten days in dry-ice-supported thermal shipping containers. After shipping, the vaccine can be stored at ultra-cold temperatures (-60°C to -90°C) but only for two weeks. After this period, the vaccine must be stored in refrigerator conditions (2°C to 8°C) for a maximum of 31 days [9]. The Moderna vaccine is, indeed, stable at (-15°C to -50°C) until its expiration date, but unused vaccines last for up to 30 days in a refrigerator [10]. In any case, ultra-cold freezing is never recommended as a long-term storage method. Instead, refrigeration offers a longer storage term with no risk of reduced potency. Though not entirely attributed to temperature abuse or damage, SARS-CoV-2 vaccines are often thrown out by the tens of thousands due to breakage, storage and transportation problems, and expiration [11]. A cold storage option covering all three conditional needs (thermal shipping, ultra-cold storage, refrigeration upon thawing) can significantly benefit the vaccination effort against COVID-19 by boosting efficiency and distribution simplicity.”
- The authors tested the longevity of this method for 7 days only. Could the authors provide additional data about longer vaccine storage using this method?
The presented study was at the experimental stage for proof of concept. Preliminary results based on repeated 7 day preservation trials seem promising in applying the technology to prevent vaccine freezing associated with temperature abuse or shock-induced circumstances during storage and transportation. In short, there will be no technical hurdle to extend our vaccine storing periods to weeks or months.
- Finally, could the authors comment on the potential application of this method to cryopreserve mammalian cells?
The following paragraphs were added in the Conclusion.
“In addtion, cryopreservation has been successful in numerous cell types and some simple tissues. However, conventional approaches to cryopreservation cannot be applied to more complex natural, or engineered, multicellular tissues due to the destructive effect of extracellular ice formation. Restricting the size and extent of ice crystal formation during cryopreservation was marginally achieved using sufficiently high concentrations of cryoprotectants to promote amorphous solidification (vitrification) rather than crystallization. The developed supercooling technology is expected to be an alternative to cryopreservation technology for mammalian cells, organs, and tissues by lessening the toxicity issue of cryoprotectants and simplifying the procedure.”

Reviewer 2 Report
The manuscript is scientifically interesting and with important applicability implications in the future methods of vaccines distribution end storage. Further investigation and testing would be extremely interesting.
Author Response
The manuscript is scientifically interesting and with important applicability implications in the future methods of vaccines distribution end storage. Further investigation and testing would be extremely interesting.
The authors appreciated it for the reviewer’s support and encouragement.

Reviewer 3 Report
The presented manuscript is well written and discussed. Thanks to the author presenting a very important topic in the filed of vaccines.
Author Response
The presented manuscript is well written and discussed. Thanks to the author presenting a very important topic in the field of vaccines.
The authors appreciated it for the reviewer’s support and encouragement.

Reviewer 4 Report
In the manuscript, Jun sun et al, to test the OMF-based supercooling technology in prevention of freeze damages of vaccines at subzero temperatures by comparing results with the aforementioned shake test and light microscopy.
The aim of the study is clear. The experiments are well designed and properly controlled. However, the reviewer have few concerns as below.
Major comments.
- Do the more numbers and different types of vaccines to support for the present study and conclusion.
Minor comments.
- Include a flow chart for study design.
Author Response
In the manuscript, Jun sun et al, to test the OMF-based supercooling technology in prevention of freeze damages of vaccines at subzero temperatures by comparing results with the aforementioned shake test and light microscopy. The aim of the study is clear. The experiments are well designed and properly controlled. However, the reviewer has a few concerns as below.
Major comments.
- Do more numbers and different types of vaccines to support for the present study and conclusion.
As a proof of concept (POC) trial, this study was intended to test and optimize the developed supercooling protocols for the prevention of potential freeze damages on the adsorbed vaccine at subzero temperature. The authors agreed that we will commit a follow-up study to examine the developed technology for diverse vaccines and quantities.
Minor comments.
- Include a flow chart for the study design.
The flow chart was created and added as the reviewer pointed out.
